# Screening of Different Essential Oils Based on Their Physicochemical and Microbiological Properties to Preserve Red Fruits and Improve Their Shelf Life

**DOI:** 10.3390/foods12020332

**Published:** 2023-01-10

**Authors:** Ziba Najmi, Alessandro Calogero Scalia, Elvira De Giglio, Stefania Cometa, Andrea Cochis, Antonio Colasanto, Monica Locatelli, Jean Daniel Coisson, Marcello Iriti, Lisa Vallone, Lia Rimondini

**Affiliations:** 1Department of Health Sciences, Center for Translational Research on Autoimmune and Allergic Diseases—CAAD, Università del Piemonte Orientale UPO, Corso Trieste 15/A, 28100 Novara, Italy; 2Department of Chemistry, University of Bari, Via Orabona 4, 70126 Bari, Italy; 3National Consortium of Materials Science and Technology (INSTM), Via G. Giusti 9, 50121 Florence, Italy; 4Jaber Innovation s.r.l., Via Calcutta 8, 00144 Rome, Italy; 5Department of Pharmaceutical Sciences, Università del Piemonte Orientale, Largo Donegani 2, 28100 Novara, Italy; 6Department of Biomedical, Surgical and Dental Sciences, Università degli Studi di Milano, Via Cesare Saldini 50, 20133 Milano, Italy; 7Department of Veterinary Medicine and Animal Sciences, Università degli Studi di Milano, Via dell’Università 6, 26900 Lodi, Italy

**Keywords:** essential oils, perishable fruits, fungicide, shelf life, physicochemical property

## Abstract

Strawberries and raspberries are susceptible to physiological and biological damage. Due to the consumer concern about using pesticides to control fruit rot, recent attention has been drawn to essential oils. Microbiological activity evaluations of different concentrations of tested EOs (cinnamon, clove, bergamot, rosemary and lemon; 10% DMSO-PBS solution was used as a diluent) against fruit rot fungal strains and a fruit-born human pathogen (*Escherichia coli*) indicated that the highest inhibition halos was found for pure cinnamon and clove oils; according to GC-MS analysis, these activities were due to the high level of the bioactive compounds cinnamaldehyde (54.5%) in cinnamon oil and eugenol (83%) in clove oil. Moreover, thermogravimetric evaluation showed they were thermally stable, with temperature peak of 232.0 °C for cinnamon and 200.6/234.9 °C for clove oils. Antibacterial activity evaluations of all tested EOs at concentrations from 5–50% (*v*/*v*) revealed a concentration of 10% (*v*/*v*) to be the minimum inhibitory concentration and minimum bactericidal concentration. The physicochemical analysis of fruits in an in vivo assay indicated that used filter papers doped with 10% (*v*/*v*) of cinnamon oil (stuck into the lids of plastic containers) were able to increase the total polyphenols and antioxidant activity in strawberries after four days, with it being easier to preserve strawberries than raspberries.

## 1. Introduction

Small red fruits such as strawberry (*Fragaria × ananassa*) and raspberry (*Rubus idaeus* L.) are considered healthy fruits thanks to their high content in terms of health-beneficial bioactive compounds such as vitamins (vitamins A, C and E), β-carotene, phenolic (flavonoids and anthocyanin) and minerals (calcium, phosphorous, iron, potassium, sodium, magnesium, selenium and zinc), dietary fiber and antioxidants [1,2,3]. Their color is strictly dependent on the high level of anthocyanin, an important polyphenolic compound which possesses antioxidant, anti-inflammatory and anti-hyperlipidemic properties [4,5]. Therefore, it is believed that fresh fruit consumption increases the human body’s protection from various non-communicable diseases such as neurological diseases, cardiovascular disease, diabetes mellitus obesity and certain cancers [6]. Huang et al. (2016) reported that strawberries include a high volume of anti-inflammatory polyphenols which have been shown to attenuate meal-induced postprandial inflammation and oxidative stress [7]. 

The main concern related to these fruits is their short shelf life (one–two days at room temperature), low resistance to mechanical forces, excessive texture softening and susceptibility to fungal infection, which may take place before or after harvesting and during the storage steps, causing a considerable reduction in the quality of fruits [3,8]. The fungal strains frequently involved in fruit rot are *Botrytis cinerea*, *Mucor* sp., *Rhizopus nigricans* (*R. stolonifera*), *Penicillium* sp. and *Colletotrichum* sp. In particular, *B. cinerea*, known as gray mold, is a major cause of red fruit rot; it can colonize on fruit before harvest and remain latent until the environmental conditions, including the humidity, temperature and availability of nutrients, become suitable for disease development [9,10].

In addition to the economic disadvantages and financial losses created by the spoilage of fresh fruits, epidemiological surveys and occasional outbreaks demonstrate that these agricultural products can be a vector of certain human pathogens such as *Salmonella* sp., *Escherichia* sp., *Shigella* sp. and *Listeria* sp., that may enter into contact with the fruit during the pre-harvest (in the field) and post-harvest stages [11]. The main source of fruit and vegetables contamination in the field is the usage of polluted water source by human and animal sewers for irrigation [12]. One research on spinach plants reported that the contamination of this plant by the pathogen *E. coli* O157:H7 occurred more often through polluted water used in irrigation systems than polluted soil. Most pathogens can survive on a plant’s surface for a long time and colonize the internal structures of the plant [13]. 

The most common approach to address post-harvest decay and contamination, and expand the shelf life of fruit, is a combination of pesticides (during harvest) and storage at low temperatures (post-harvest process) [14]. Considering the use of pesticides on human health [15], much research has recently been carried out to seek alternative, non-damaging and eco-friendly methods for the preservation of small red fruits such as strawberries and raspberries. Among these techniques, modified atmosphere packaging [16], osmotic treatments [17], radiation and heat treatment [14], the application of plant natural products, such as essential oils, can be highlighted for the purpose of controlling fruit decay [18].

Plant essential oils (EOs) are health-beneficial bioactive compounds that are naturally responsible for the protection of plants against certain pathogenic microorganisms, undesirable insects and herbivores via the production of unpleasant tastes and smells. Terpenes (monoterpenes and sesquiterpenes), terpenoids (tymol, carvacrol, linalyl acetate and citronellal) and aromatic compounds with various biological features are the principal components found in EOs and are noticeably influenced by plant species, climate condition, geographic region, the harvesting process and the soil type [18,19,20]; additionally, their activity is not attributable to a unique mechanism but triggers a cascade of responses involving whole bacterial and fungal cells, in turn causing the inhibition of bacterial and fungal growth, conidia germination and the production of bacterial and fungal metabolites [21]. Several studies have been carried out on the antimicrobial and antifungal activity of EOs. The EOs of *Artemisia afra*, *Pteronia incana* and *Rosmarinus officinalis* showed a broad spectrum of antimicrobial activity on 41 microbial strains including food spoilage as well as human and plant pathogenic bacterial and yeast strains [22]. The EOs of hyssop were shown to inhibit plant pathogenic fungi such as *Pyrenophora avenae* and *Pyricularia oryzae* and inhibit the germination of conidia of *Botrytis faba* [23]. Research has been carried out to investigate the effect of the EOs on fruit decay. For instance, Ansarifar et al. (2021) investigated quality preservation of strawberries using thyme EO encapsulated in zein nanofiber. Since the thyme oil was gradually released into the fruits’ packages, their physicochemical properties remained constant and bacterial and fungal contaminations decreased after 15-day storage at 4 °C [24]. In addition, Guerreiro et al. (2021) reported that using an edible coating with a combination of alginate (2%), citral (0.15%) and eugenol (0.1%) preserved strawberries with no changes in sensorial and nutritional properties for up to 15 days [25]. However, there are few reports assessing the expansion of fresh fruits by simulating real packaging conditions in fields and the market place at room temperature. Arroyo et al. (2007) superficially inoculated strawberries with a conidial suspension of *Candida acutatum* (10^6^ conidia/mL), and the fruits were placed tightly on pieces of filter paper doped with different doses of EO compounds [26]. Wang et al. (2007) analyzed changes in strawberry extracts after placing them into a 1L polystyrene containers with lids including 200 mg of different EO compounds such as thymol, eugenol and menthol inside small bakers [27]. However, the aim of this study was the investigation of the EOs’ effects on the shelf life of fruits without external and prior bacterial or fungal infection. This current work represents a comprehensive piece of research including the physicochemical analysis of EOs, their biological activity and in vivo evaluation, along with the simulation of real packaging. The inclusion of these data in one study is uncommon in previous publications because they have tended to focus on a specific topic. 

A report by the Food and Drug Administration (FDA) published in 2020 indicated 48 million cases of foodborne illness yearly in the United States, with an estimated 128,000 hospitalizations and 3000 deaths. The common bacterial strains responsible for foodborne outbreaks are as follows: *E. coli*, *Salmonella* sp. and *Listeria* sp. [28]. Therefore, in this study, as in previous studies in the literature [29,30], *E. coli* was selected as a representative of human foodborne pathogens.

First, the characterization of EOs using Fourier-Transform Infrared Spectroscopy in Total Attenuated Reflectance mode (FTIR/ATR), as Chromatography -Mass Spectrometry (GC-MS), High- Performance Liquid Chromatography (HPLC) and Thermogravimetric Analysis (TGA) was assessed; then, their antifungal and antibacterial activities against a wide range of fungal strains responsible for fruit and vegetable spoilage and *E. coli*, as a common human pathogen transmitted through these agricultural products, were investigated. For prospective experiments, the antibacterial activities of binary combined EOs were analyzed, and the interactions between EOs was detected. As a final step, the highest potential EO was selected for in vivo experiments, and, after four days at room temperature, analyses regarding the quality of fruits were performed. 

## 2. Materials and Methods

### 2.1. Preparation of Samples

The essential oils (cinnamon, clove, rosemary, lemon and bergamot) used for this study were commercially purchased from Farmalabor (Milan, Italy). The pure EOs were considered as 100%, and, in order to prepare different tested concentrations (10%, 1% and 0.1% *v*/*v*), 100, 10 and 1 µL of pure EOs were dissolved in 900, 990 and 999 µL of 10% Dimethyl Sulfoxide (DMSO, Merck, Milan, Italy) in Phosphate Buffer Saline (PBS, Sigma-Aldrich, Milan, Italy) solution, respectively. These concentrations of EOs were chosen according to previous literature indicating the cytotoxic effects of pure EOs; for instance, Azzimonti et al. (2015) investigated Human Gingival Fibroblast (HGF) and Mucosal Keratinocyte (HKF) viability at three different dilutions of EO (10-, 100- and 1000-fold). Their results revealed an approximately 40% reduction in HGF and HKF viability at a dilution of 10-fold in compared to 100- and 1000-fold dilutions [31]. 

Pure and diluted EOs were stored protected from light at 4 °C. All the procedures were carried out under a laminar air flow cabinet to preserve sterile conditions. Before each experiment, the prepared dilutions were vigorously vortexed (1 min, room temperature). For all experiments, a solution of 10% DMSO in PBS was used as a control sample.

### 2.2. Chemical Characterization of Essential Oils

#### 2.2.1. Chromatographic Analyses 

Bergamot, lemon, rosemary and cinnamon EOs were quantified by means of Gas Chromatography Mass Spectrometry (GC–MS), as previously described by Pinto et al. [32], albeit with certain modifications. GC–MS analyses were performed using a gas chromatograph 680 coupled with a Clarus SQ 8 T mass spectrometer (PerkinElmer Inc., Waltham, MA, USA). The gas chromatograph was equipped with a split/split-less inlet, an ELITE 5-MS column (PerkinElmer Inc., Waltham, MA, USA) (30 m length × 0.25 mm inner diameter × 0.25 μm full thickness) and helium (48 kPa~7 psi) as the carrier gas, at a constant flow rate of 1.2 mL/min. The EO (0.1%) was injected (1 μL) into the column at a split ratio of 2; the injector temperature was 250 °C. The dilution of EOs was carried out in methanol for cinnamon oil and in hexane for the other oils. The GC-MS ion source temperature was set to 250 °C. The Total Ion Current (TIC) chromatograms were taken under positive electron impact ionization mode, with a mass range from 55 to 210 amu, a solvent delay of 5 min and a transfer line of 275 °C. The chromatographic program was for bergamot: 50 °C for 2 min, a gradient of 30 °C/min until 240 °C and then 240 °C for 2 min; lemon: 50 °C for 2 min, a gradient of 25 °C/min until 240 °C and then 240 °C for 2 min; rosemary: 55 °C for 2 min, a gradient of 25 °C/min until 260 °C and then 260 °C for 2 min; and cinnamon: 50 °C for 2 min, a gradient of 25 °C/min until 260 °C and then 260 °C for 2 min. The concentration of each compound was calculated by means of area interpolation on the calibration curve built using external analytical standards (Sigma-Aldrich Srl, Milan, Italy).

As far as clove oil detection is concerned, HPLC monitoring of eugenol (the main component of clove oil) was carried out, employing HPLC (Prominence Series 20 with SPD-M20A PDA detector, Shimadzu, Milan, Italy) by adapting the method previously described by Yun et al. [33]. A Shim-Pack GIST C18-AQ column (150 mm × 4.6 mm, 5 µm Shimadzu, Milan, Italy) was eluted in isocratic mode at 30 °C, 40% methanol and 60% water. The effluent was monitored at 215 nm. The mobile phase flow rate was kept at 1 mL/min, and samples were injected through a 20 µL injection loop. LabSolutions software (for HPLC Prominence Series 20 with SPD-M20A PDA detector, Shimadzu, Milan, Italy) was exploited to build a calibration curve (r^2^ 0.999) with the standard compound dissolved in the mobile phase at four concentrations (1, 5, 10, 25 µg/mL). Each sample was tested in triplicate, and data were reported as mean ± standard deviation.

#### 2.2.2. Fourier-Transform Infrared Spectroscopy 

Fourier-Transform Infrared/Attenuated Total Reflection (FTIR/ATR) analyses was performed using a Spectrum Two-PE instrument endowed with a universal ATR accessory (UATR, Single Reflection Diamond/ZnSe) supplied by PerkinElmer Inc. (Waltham, MA, USA). For each of the relevant EOs, FTIR/ATR spectra were recorded from 400 to 4000 cm^−1^ with a 4 cm^−1^ resolution, with the liquids being placed directly over the diamond.

#### 2.2.3. Thermogravimetric Analysis 

Thermogravimetric (TGA) is a widely used technique to evaluate the thermal stability of materials in which the loss of mass as a function of temperature is measured. EOs were examined through a PerkinElmer TGA-400 instrument (PerkinElmer Inc., Waltham, MA, USA). Briefly, 5–10 mg samples were heated in an air-saturated atmosphere in the range of 30–800 °C, with a constant flow rate (20 °C/min) and a gas flow set at 20 mL/min. The TGA Pyris series software (PerkinElmer Inc., Waltham, MA, USA) was exploited to record Thermograms (TG) and calculate their Respective Derivative Curves (DTG) for further data mining.

### 2.3. Antifungal and Antibacterial Activity Evaluation

#### 2.3.1. Strains

##### Fungal Strains

Fungal strains that isolated wildly from spoiled fruits and vegetables were included: *Aspergillus terreus* (from rye), *Penicillium italicum* (from citrus fruits), *Fusarium verticillioides* (from maize), *Byssochlamys nivea* (from canned fruits and juices) and *Rhizopus nigricans* (from strawberry). Briefly, the spoiled fruits were cut into small pieces with a sterile blade and plated onto Sabouraud Dextrose Agar (SDA, Merck; Milan, Italy) for 5 days at 28 °C. Different fungal colonies appeared and were characterized visually through their morphology under a light digital microscope. Pure fungal strains were obtained by the sub-culturing each of the selected colonies in Sabouraud Dextrose Broth (SDB) or M2 Broth (Merck, Milan, Italy). *B. cincerea* was purchased from the Mycotheca Universitatis Taurinensis (MUT00003006; Turin, Italy) and sub-cultured in SDB or M2 broth for 5 days at 28 °C. A fresh broth culture was prepared prior to each experiment; the fungal concentration was adjusted to 1 × 10^4^ spores/mL by diluting spores counted with a Burker chamber in the fresh media. All the procedures were carried out under a laminar air flow cabinet to provide sterile conditions.

##### Bacterial Strain

*Escherichia coli* (ATCC 25922, Gram-negative) was purchased from the American Type Culture Collection (ATCC, Manassas, VA, USA). Bacterium was cultivated on Trypticase Soy Agar (TSA, Merck, Milan, Italy) and incubated at 37 °C until round single colonies were formed; then, 2–3 colonies were collected and spotted into 30 mL of Luria Bertani (LB, Merck, Milan, Italy) broth. The broth culture was incubated overnight at 37 °C under agitation (120 rpm in an orbital shaker). A fresh culture was prepared prior to each experiment; the bacterial concentration was adjusted to 1 × 10^5^ cells/mL by diluting in the fresh media until an optical density of 0.001 at 600 nm was reached as determined by a spectrophotometer (Spark, Tecan, Switzerland). LB was used as a blank. All the procedures were carried out under a laminar air flow cabinet to provide sterile conditions.

#### 2.3.2. Antifungal Activity of Essential Oils

An investigation of antifungal activity of EOs was performed by disc diffusion according to the International Organization for Standardization (ISO 16782, 2016) [34] and Clinical and Laboratory Standards Institute (CLSI M02-A10) [35]. Briefly, SDA were streaked with 1 mL of each fungal strain, *A. terreus*, *P. italicum*, *F. verticillioides*, *B. nivea*, *R. nigricans* and *B. cinerea*, at a concentration 1 × 10^4^ spores/mL, and then left under the laminar air flow cabinet to be properly dried. The experiment was followed by placing blank filter paper discs (6 mm in diameter) sterilized by UV irradiation (2 h in 5 cm distance) on the middle of the infected SDA plates. Then, 20 µL of each EO concentration (pure oils, 10%, 1% and 0.1% *v*/*v* detailed in the Section 2.1) was added onto blank discs, which were kept under the laminar air flow cabinet for 30 min until the EOs were completely absorbed into the discs. Finally, they were incubated for 5 days at 28 °C; then, the diameter of inhibition halo created around the disc was measured, which indicates the efficacy of the antifungal activity of EOs.

#### 2.3.3. Antibacterial Activity of Essential Oils

Investigation of antibacterial activity of EOs was performed by both techniques including disc diffusion and direct assay; In direct way, *E. coli* (Gram-negative bacterial strain) suspension in LB broth was exposed to different concentrations of EOs (pure oils, 10%, 1% and 0.1% *v*/*v* detailed in the Section 2.1) in a 48 multi-well plate. Briefly, in each well of the multi-well plate a volume of 300 µL containing 200 µL of *E. coli*, at concentration 1 × 10^5^ cells/mL and 100 µL of different concentrations of EOs (pure, 10%, 1% and 0.1% *v*/*v*) was added. After 24 h bacterial metabolic activity was evaluated by the resazurin colorimetric metabolic assay (alamarBlue^TM^, ready-to-use solution from Life Technologies, Milan, Italy) by directly adding the dye solution (0.0015% in PBS) onto the infected specimens. After 1 h incubation in the dark the fluorescent signals (expressed as relative fluorescent units—RFU) were detected at 590 nm by spectrophotometer (Spark, Tecan, Switzerland). To carry out disc diffusion, LB agar plates were streaked with 1 mL of LB containing bacterial strain at concentration 1 × 10^5^ cells/mL and followed with the same process as antifungal process as above-mentioned (Section 2.3.2). After incubation for 24 h at 37 °C, the diameter of the inhibition halo around the disc was measured, which indicates the efficacy of antibacterial activity of EOs. 

#### 2.3.4. Determination of Minimum Inhibitory Concentration (MIC) and Minimum Bactericidal Concentration (MBC) for Bacterium

A wide range of concentrations of EOs (pure, 50%, 25%, 16.6%, 12.5%, 10% and 5% (*v*/*v*)) against *E. coli* was evaluated to determine the Minimum Inhibitory Concentration (MIC) by the microdilution broth method (CLSI M07-A10, 2009) [35]. A total of 200 µL of *E. coli* at a concentration of 1 × 10^5^ cells/mL was exposed to 100 µL of each concentration of EOs in a 48 multi-wells plate and incubated at 37 °C for 24 h. The optical density of bacteria was measured at a wavelength of 600 nm by a spectrophotometer indicating the bacterial growth, and the minimum EO concentrations inhibiting the bacterial growth were considered as MICs.

To investigate the Minimum Bactericidal Concentration (MBC), from each concentration used for the MIC assay, 200 µL was transferred on an LB agar plate and incubated at 37 °C for 24 h. The results were visually checked to determine the lowest bactericidal concentration of EOs as MBC. 

### 2.4. Interaction between Essential Oils

To investigate whether EOs have an impact on each other, a binary combination of them at two concentrations, 5% and 10% (*v*/*v*), was performed, and Table 1 represents the combination conditions. Then, the effect of the combined EOs against *E. coli* at a concentration 1 × 10^5^ cells/mL was compared with the effect of individual oil through the disc diffusion assay [35]. After 24 h of incubation and the measurement of the inhibition halo diameter, a Fractional Inhibitory Concentration (FIC) index indicating the interaction between EOs was calculated as follows (Equation (1)):FIC index = FIC_A_ + FIC_B_
FIC_A_ = MIC (A in the presence of B)/MIC (A alone)
FIC_B_ = MIC (B in the presence of A)/MIC (B alone)(1)

An FIC index of <0.5 indicates synergism, a 0.5 < FIC index < 1 indicates additive, a 1 < FIC index < 4 determines indifference and an FIC index > 4 determines antagonism [36].

### 2.5. Impact of Essential Oils on Shelf Life of Fruits

To find out whether EOs can increase the shelf life of small red fruits, biological strawberries and raspberries were purchased from an organic field where no chemical pesticides had been used for plantation. According to the obtained results, cinnamon oil at a concentration of 10% (*v*/*v*) was selected for an in vivo experiment. In brief, filter paper at the dimension 3 × 3 cm sterilized by UV irradiation (for 2 h at a distance of 5 cm) were completely covered by adding dropwise 120 µL of cinnamon oil at a concentration of 10% (*v*/*v*), with these being stuck to the inner part of a sterile plastic container’s lid (dimension of the plastic container is 4 × 6 cm). Then, fruits of identical size, shape and texture were chosen and divided into two groups: (1) including filter paper loaded with cinnamon oil at a concentration of 10% (*v*/*v*); (2) including filter paper loaded with 10% DMSO in PBS solution, as a control sample. Based on the fruits’ sizes and the plastic container space, two strawberries and three raspberries were put in each plastic container to replicate. All these procedures were performed under the laminar air flow cabinet to provide sterile conditions. After preparation, they were incubated at room temperature for 5 days; the morphologies of fruits were analyzed visually each day and their physicochemical properties were assessed every two days. 

### 2.6. Physicochemical Characterization of Fruits

At each time point, fruits were homogenized using an Ultraturrax (TP 18–10, IKA Werke, Germany) at 13,500 rpm for 2 min; then, their physicochemical properties, such as their weight loss, moisture values, total soluble solids, pH and titratable acidity, were determined as follows: 

#### 2.6.1. Moisture Values

The moisture values were obtained by using a Sartorius MA35 Moisture Analyzer (Goettingen, Germany).

#### 2.6.2. Total Soluble Solids

To measure the total soluble solids, first, homogenized fruits were centrifuged at 5000 rpm for 20 min at 4 °C to remove insoluble materials from the sample. Then, Brix degrees were measured with an Atago refractometer at 20 °C (Tokyo, Japan) [37,38].

#### 2.6.3. Maturation Index

The maturation index (MI) was calculated as the ratio between soluble solids and acidity (Equation (2)) [39]:MI = BRIX/ACID(2)

#### 2.6.4. pH and Titratable Acidity

The pH was measured with a pH meter (Mettler-Toledo, Columbus, OH, USA) after calibration with buffer solutions (pH 4, 7 and 10). The titratable acidity was performed by titrating 5 mL of homogenized fruits dispersed in 50 mL of distilled water using 0.1 N NaOH and phenolphthalein as the indicator. The results were expressed as the % citric acid and calculated using Equation (3):% Citric acid = [(V1 × N)/V2] × K × 100(3)
where V1 is the volume of NaOH consumed (mL), V2 is the sample volume (mL), K is the equivalent weight of the citric acid (0.064 g/meq) and N is the normality of NaOH (0.1 meq/mL) [39].

#### 2.6.5. Preparation of the Phenolics Extract

Samples (2.5 g) were added to 5 mL of ethanol/water (70/30) solution and shaken vigorously by a vortex (VWR Analog Vortex Mixer, PA, USA). The samples were then subjected to ultrasound extraction (Branson 1510, CT, USA) at room temperature for 30 min. Finally, the extract was centrifuged for 10 min at 9200× *g* and 4 °C (Eppendorf Centrifuge 5417R, Hamburg, Germany). The extracts were immediately tested for the following analyses: antioxidant activity (Section 2.6.6), total phenolic content (Section 2.6.7) and total and monomeric anthocyanins (Section 2.6.8).

#### 2.6.6. Antioxidant Activity

Antioxidant activity was carried out by using the 1,1-diphenyl-2-picrylhydrazyl (DPPH) radical scavenging assay [40]. DPPH radical is violet/purple colored in methanol solution and turns into a yellowish color in the presence of antioxidants. In brief, 700 µL of phenolic extract diluted 1:500 with methanol was added to the same volume of 100 µM methanolic solution of DPPH. The reaction mixture was shaken completely and left in the dark at room temperature for 20 min [40]. The absorbance of the mixture was measured with a Shimadzu UV-1900 spectrophotometer (Shimadzu Italia, Milan, Italy) at 515 nm, and the DPPH inhibition percentage was calculated. The DPPH radical scavenging activity was expressed as mg of Trolox equivalent/Kg of sample (d.w.) through a calibration curve. All tests were performed in triplicate [40].

#### 2.6.7. Total Phenolic Content

The total phenolic content was determined using modified Folin–Ciocalteu methods. Briefly, 50 µL of Folin–Ciocalteu reagent and 175 µL of aqueous Na_2_CO_3_ (5% *w*/*v*) were added to 50 µL of phenolic extract. The solution was then diluted to a final volume of 2900 µL with distilled water. After 1 h, the absorbance was measured with the spectrophotometer (Shimadzu UV-1900, Milan, Italy) at 760 nm [41]. 

#### 2.6.8. Total and Monomeric Anthocyanin

The quantification method for Total Anthocyanins (TA) and Total Monomeric Anthocyanins (TMA) was carried out according to Colasanto et al. [42]. Phenolic extract was opportunely diluted with potassium chloride buffer (0.025 M), pH 1.0, until the absorbance of the sample at 520 nm was within the linear range of the spectrophotometer. The same Dilution Factor (DF) was applied to the dilution with sodium acetate buffer (0.4 M), pH 4.5. Solutions at pH 1.0 were let to equilibrate for 5 min and those at pH 4.5 for 15 min; then, the absorbance was measured at both 520 and 700 nm. The concentrations of TA and TMA in the extracts were expressed as cyanidin-3-O-glucoside (Cy-3-Glu) equivalents according to the Equations (4) and (5):TA (µg/mL) = [[(A520 nm − A700 nm) × 449.2 × DF × 1000]/(Ɛ × 1)] × 100(4)
TMA (µg/mL) = [[(A520 nm − A700 nm)pH 1.0 − (A520 nm − A700 nm)pH 4.5 × 449.2 × DF × 1000]/(ε × 1)] × 100(5)
where 449.2 is molecular weight of Cy-3-Glu, 1000 is conversion factor from g to mg and ε (molar extinction coefficient of Cy-3-Glu) is 26,900 L mol^−1^ [43]. The results were expressed as mg of Cy-3Glu equivalents (C3GE) for g of dry matter.

### 2.7. Statistical Analysis of Data 

Experiments were performed in triplicate. Results were statistically analyzed using the SPSS software (v.20.0, IBM, New York, NY, USA). First, data normal distribution and homogeneity of variance were confirmed by the Shapiro–Wilk and the Levene test, respectively; then, groups were compared by using the one-way ANOVA using the Tukey’s test as post-hoc analysis. Significant differences were established at *p* < 0.05.

## 3. Results

### 3.1. Characterizations of Essential Oils

The EOs purchased from Farmalabor had been analyzed by the company, but in the technical datasheet only a range of relative abundances were reported. Therefore, the analysis of the Eos’ compositions was carried out to ascertain compositions that may change from batch to batch. The main components of the investigated EOs (bergamot, lemon, rosemary and cinnamon), approximately 71–79% of their compositions, are reported in Table 2, indicating that limonene (39.5% and 63.7% in bergamot and lemon oils, respectively), eucalyptol (60.5% in rosemary), cinnamaldehyde (54.5% in cinnamon oil) were the dominant compositions. The missing percentages (about 20–30%) were due to other components showing relative abundances for each molecule <5%. 

The main components detected in our investigation were almost similar to those reported in the literature for bergamot [44], lemon [45,46], rosemary [47] and cinnamon oils [48]. However, it is important to underline that different relative abundances between our results and those reported in the literature were expected since several factors, such as geographic areas, extraction methods, etc., significantly influence EO composition. Finally, clove oil was analyzed by means of HPLC, evidencing a eugenol amount equal to 83%.

FTIR/ATR spectra of the investigated EOs (see Appendix A) showed, in the primary cases, the typical bands in terms of aromatic CH bonds, between 3100 and 3000 cm^−1^, CH present in alkenes, between 3080 and 3020 cm^−1^, and C=C stretching vibration, between 1680 and 1640 cm^−1^. In addition, cinnamon oil showed an intense band relevant to aldehyde C=O of cinnamaldehyde at 1672.4 cm^−1^. A C=O stretching band was found in rosemary oil, due to the cheton group of camphor (1746.2 cm^−1^), and in bergamot oil, due to the ester group (1734.0 cm^−1^) of linalyl acetate. On the other hand, lemon and bergamot oils showed typical absorption bands in terms of terpenes and terpenoids. Clove oil spectrum results were very similar to that of eugenol (data not shown), in agreement with the high eugenol percentage present in this EO.

Moreover, in view of a potential employment in food packaging, thermogravimetric analyses of the EOs were carried out in order to gain insight into their vaporization behavior and thermal stability. This technique has been widely used to study the thermal properties of EOs [49,50]. EOs are usually highly sensitive to environmental conditions, such as temperature, light and oxygen. Therefore, knowledge of the thermal stability of EOs is fundamental. In Figure 1, the thermogravimetric and (in the inset) the derivative traces of bergamot, lemon, rosemary, clove and cinnamon oils are reported. Additionally, in Table 3, the onset (calculated considering 1% of weight loss) and peak temperatures of all the investigated oils are reported. According to these results, it can be concluded that the EOs were thermally stable and their temperature peak was within the range of 131.6/158.6 °C for lemon oil to 232.0 °C for cinnamon oil. 

The thermogravimetric analysis evidenced the high volatile behavior of citrusy EOs as well as rosemary oil. In particular, considering the onset temperature values, lemon oil was the least thermally stable, followed by rosemary oil. The cinnamon and clove oils were found to be the most thermally stable. The lemon and rosemary oils started to degrade at temperatures close to 50 °C, while the other EOs were stable. As far as cinnamon oil was concerned, the recorded T_peak_ was slightly higher than that reported for cinnamon oil analyzed by Feng et al. (2017), where the stage ended at 225 °C. They associated this stage with the decomposition of stable volatile/non-volatile components [51]. As far as clove oil is concerned, a high similarity between the EO and eugenol thermograms was found. Indeed, similar to the case of clove oil, eugenol degraded in two temperature stages, i.e., 209 and 235 °C (data not shown). This finding is in agreement with the literature [52] and was supported by our HPLC results, which evidenced a high percentage of eugenol in clove oil. 

### 3.2. Antifungal Screening of Essential Oils

The fungal strains used in this study, including *Aspergillus terreus*, *Penicillium italicum*, *Fusarium verticillioides*, *Byssochlamys nivea* and *Rhizopus nigricans*, were isolated directly from spoiled fruits and vegetables. Figure 2 presents the results of the antifungal activity of EOs through the measurement of diameter of inhibition halos around the discs. All tested fungal strains showed sensitivity in the presence of pure cinnamon oil; the most affected fungal strains were *A. terreus* and *B. nivea*, with inhibition halos of approximately 8 and 7 cm, respectively. Additionally, *P. italicum* and *F. verticillioides* remained sensitive when they were exposed to low concentrations of cinnamon oil (1% and 0.1% *v*/*v*). The second EO with high potential of antifungal activity was clove oil. Indeed, all fungal strains used in this study revealed sensitivity to clove pure oil. The highest effect was shown against *F. verticillioides* and *A. terreus*, with inhibition halos ~6 cm; moreover, *F. verticillioides*, *A. terreus*, *R. nigricans* and *P. italicum* remained sensitive at a concentration of 10% (*v*/*v*) of clove oil, with inhibition halos of 4.5, 3.5, 3 and 1 cm, respectively (Figure 2). The antifungal activity of other EOs, including bergamot, rosemary and lemon, were observed only in pure concentrations as follows: *B. cinerea* showed sensitivity in the presence of all three of these EOs, with a range of inhibition halos between 1–2 cm; *F. verticillioides* was sensitive in the presence of bergamot and lemon oils, with inhibition halos of 4.5 and 2 cm, respectively; *A. terreus* was sensitive in the presence of lemon and rosemary oils, with inhibition halos of 4 and 1 cm, respectively; and *R. nigricans* was sensitive in the presence of rosemary oil, with an inhibition halo of 3 cm (Figure 2). 

Indeed, the high potential activity of cinnamon oil employed in this work against fungal strains is probably due to its large amounts of bioactive components such as cinnamaldehyde and caryophyllene, plus other terpenic species, in particular high quantity of eugenol, that were detected by GC-MS analysis (see Table 2). These obtained results are in agreement with previous research indicating that cinnamon oil has been determined to be Generally Recognized as Safe (GRAS) as a food ingredient due to its bioactive compounds [53]. Xing et al. (2010) demonstrated the antifungal activity of cinnamon oil against *Rhizopus nigricans*, *Aspergillus flavus* and *Penicillium expansum* at an MIC of 0.64%, 0.16% and 0.16% (*v*/*v*), respectively [54]. Hu et al. (2019) investigated the antifungal activity of different EOs on three common fungi isolated from moldy wheat bread, namely *A. niger*, *A. oryzae* and *A. ochraceus*; among the tested EOs, cinnamon oil showed the highest antifungal activity [55]. He et al. (2018) reported that *Colletotrichum acutatum* isolated from decayed Kiwi was significantly sensitive when exposed to cinnamon oil at a concentration of 0.2 µL/mL due to loss of integrity of the cell membrane [56]. 

The results indicate that, in addition to cinnamon oil, the other EO with considerable efficiency in terms of antifungal activity is clove oil; according to HPLC analysis, its biological activity was due to the presence of a high level of the bioactive component eugenol (approximately 83% of its composition, see Section 3.1). The obtained results are in agreement with the literature. Singh Rana (2011) demonstrated the antifungal activity of clove oil on common fungal pathogens of plants and animals such as *F. moniliforme*, *F. oxysporum*, *Aspergillus* spp., *Mucor* spp., *Trichophyton rubrum* and *Microsporum gypseum* [57].

### 3.3. Antibacterial Screening of Essential Oils

In order to investigate the antibacterial activity of EOs, the metabolic activity of *E. coli* was measured by alamarBlue colorimetric assay and the results are presented in Figure 3. All the results were normalized with respect to culture media containing bacteria in the presence of 10% DMSO in PBS solution as a control sample. As shown in Figure 3, in addition to the pure concentration of EOs significantly reducing the metabolic activity of *E. coli*, both concentrations, 10% and 1% (*v*/*v*), of cinnamon and clove oils produced a 95% reduction in metabolic activity. Moreover, the lowest concentration, 0.1% (*v*/*v*), of cinnamon oil resulted in approximately 30% of bacterial metabolic activity, while the same concentration of clove oil resulted in 70% of metabolic activity. Considering the other EOs (lemon, bergamot and rosemary), a significant reduction in bacterial metabolic activity occurred at a concentration of 10% (*v*/*v*) as follows: the rosemary, lemon and bergamot oils resulted in metabolic activities of 5%, 17% and 25%, respectively. These results were confirmed by the results of disc diffusion: the inhibition halo for pure cinnamon oil was 4.4 cm and for cinnamon oil at a concentration of 10% (*v*/*v*) and pure clove oil they were 3 and 2.4 cm, respectively (Appendix A).

### 3.4. Determination of Minimum Inhibitory Concentration (MIC) and Minimum Bactericidal Concentration (MBC)

According to the obtained results for antibacterial and antifungal activity, the following concentrations of EOs were selected for investigation of MIC: 5%, 10%, 12.5%, 16.5%, 25% and 50% *v*/*v*. The results are reported in Figure 4. The minimum concentration of all the tested EOs that inhibits bacterial growth was 10% (*v*/*v*). In addition, this reduction in bacterial viability remained even, with no significant difference at concentrations over 10% (*v*/*v*) (in Figure 4, the red line indicates the Relative Fluorescent Unit (RFU) of the solution 0.00015% alamarBlue in PBS). 

In order to determine the MBC, an LB agar plate was streaked with the same EO concentrations used for the MIC experiment. As the results shown in Appendix A, only one colony of *E. coli* was observed at a concentration of 5% (*v*/*v*) of cinnamon oil; no bacterial colonies were detected at the other EO concentrations, indicating that the MBC was 5% (*v*/*v*) for clove, bergamot, lemon and rosemary oils and 10% (*v*/*v*) for cinnamon oil (Appendix A). 

These results are in agreement with previous research. Sakkas et al. (2018) investigated the antibacterial activity of different EOs including basil, chamomile, origanum, tea tree and thyme against a wide range of bacterial strains isolated from wastewater treatment plants and clinical samples. Their results showed that the MIC values for enterococci varied from 0.25 to 1% (*v*/*v*) for origanum oil, 0.5 to 2% (*v*/*v*) for thyme oil, 1 to 4% (*v*/*v*) for tea tree oil, 4 to >4% (*v*/*v*) for basil oil and >4% (*v*/*v*) for camomile oil [58]. Montironi et al. (2016) assessed the effect of EO extracted from the plant of *Minthostachys verticillata* on bovine mastitis, a disease mainly caused by *Streptococcus uberis*, indicating that the MIC and MBC values of the EO ranged from 1.56 to 12.5% and 12.5 to 25% (*v*/*v*), respectively [59].

### 3.5. Binary Interaction of Essential Oils

To evaluate the interaction between EOs, every EO at two different concentrations (5% and 10% *v*/*v*)) were combined together (the ratio of combination was 50:50; detailed in Table 4) and then their antibacterial activity and interaction with each other were investigated. The diameter of the inhibition halo ranged from 0.8 mm for the combination of cinnamon + lemon to 1.8 mm for cinnamon + bergamot, with both of them at a concentration of 10% (*v*/*v*) of each EO (Appendix A). Moreover, based on the FIC index result in Table 4, both the combinations of the EOs cinnamon + lemon (at concentrations of 10% and 5% *v*/*v*) and cinnamon + bergamot (at a concentration of 10% *v*/*v*) showed FIC indices <1, indicating the additive effect of these EOs on each other. Some previous studies have reported that the combination of EOs increased their antibacterial activity compared to single oils. Gadisa et al. (2019) combined three EOs extracted from the aerial portion of *B. cuspidate*, *B. ogadensis* and *T. schimper* in a 1:1 ratio, with their antibacterial activity on *E. coli*, *K. pneumoniae* and Methicillin-Resistant *S. aureus* (MRSA) being assessed with disc diffusion. They reported that there is a synergistic effect between the EOs of *B. cuspidate* and *T. schimpe*, with an inhibition halo 39 mm against MRSA and 28–35 mm against *E. coli* and *K. pneumonia* [60]. Soulaimani et al. (2021) reported that EOs extracted from *Thymus pallidus*, *Rosmarinus officinalis* and *Linaria maroccana* individually had moderate antibacterial activity, while the binary combination of EOs derived from *T. pallidus* and *L. maroccana* revealed synergistic effects against *E. coli* and *P. aeruginosa* and an additive effect against *K. pneumonia* [61].

### 3.6. Effect of Essential Oils on Strawberries and Raspberries in In Vivo Condition

According to the results obtained from previous sections, cinnamon oil with a concentration of 10% (*v*/*v*) was chosen for an in vivo experiment with two kinds of small red fruits: strawberries and raspberries. As explained in the Section 2.5, sterile filter paper was covered dropwise by EOs and then was stuck on the inner part of the lid of a plastic container. First of all, the samples were visually checked daily to monitor fruit spoilage and weight loss (% of fresh weight or f.w.) for up to four days. The physicochemical properties of the fruits were investigated every two days.

Figure 5 represents the results of visually monitoring for fruit spoilage. Raspberry spoilage in the plastic container including filter paper covered with a solution of 10% DMSO in PBS (control sample) started from day 1. On day 2, the fungal spoilage increased and spots of decay were also detectable on the strawberries. However, during incubation in the plastic container with filter paper covered by 10% (*v*/*v*) cinnamon oil, no fungal spoilage spots were observed. The dashed squares in Figure 5 indicate the spoilage spots. These results are in agreement with previous research; for instance, Piechowiak et al. (2022) showed that starch- and gelatin-based edible coatings doped with cinnamon oil (10%) considerably reduced the fungal growth on blueberries, 1.29 log cycles lower than those in the control specimens, after 10 days storage at 4 °C [62]. Rashid et al. (2020) investigated the storage stability of apples coated in different concentrations of cinnamon oil at 5 °C for two months; based on physicochemical analysis, microbial assay and sensory evaluation during the storage time, it was revealed that 5% of cinnamon oil prevented fruit spoilage with no variation in nutritional values in apples [63].

### 3.7. Physicochemical Properties of Fruits

Table 5 and Table 6 represent the results of the in vivo experiment with respect to the physicochemical properties of strawberries and raspberries in the presence of 1% DMSO in PBS solution (control) and 10% cinnamon oil, respectively. 

Specific classic parameters able to describe the fruit composition were determined for the purpose of evaluating the most evident variations during ripening (°Brix, acidity and ripening index). Regarding the control samples, without EO, two different behaviors can be noted between strawberries and raspberries: for the strawberries, there is an increase in moisture and a decrease in pH (Table 5), whereas for raspberries, the water content remains constant and the pH decreases only after four days (Table 6). In addition, in strawberries, no variation in terms of the titratable acidity was detectable, while the pH significantly decreased during storage in both control and treated fruits. It is likely that the titratable acidity only partially contributes to the pH of the fruits, and the effect of EO on the change in pH is limited. Concerning raspberry, certain significant differences among samples were observed with respect to titratable acidity; however, these differences are relevant primarily from statistical point of view, and the impact on fruit properties could be considered to be minimal. Considering the parameters of total soluble solids (TSS), acidity and their ratio (ripening index), there are significant differences over time in the case of soluble solids, while there are no variations regarding the acidity, which determines a difference in the ripening index over time. Comparing fruits packed with 10% (*v*/*v*) cinnamon oil and the control, only a significant difference in the maturation index is present at time T2, due to a lower TSS value in the control.

In the case of the parameters characterizing the polyphenolic fraction, it can be noted for the strawberries that there is an increase in anthocyanins (total and monomeric) with storage, without differences between controls and fruits packed with EO. Total polyphenols and antioxidant activity do not show significant variations over time, but the differences between controls and fruits packed with EO are significant, especially after four days of storage, with higher values in the samples stored in the presence of 10% cinnamon oil (Table 6).

Considering the data of raspberries, significant variations are also to be noted with regard to the ripening index, with differences in both factors (titratable acidity and TSS); even the fruits packed with EO showed a significant difference in comparison to the controls. In addition, there are no significant variations in the cases of total polyphenols and total antioxidant activity, while the evaluation of the anthocyanin content (total and monomeric) is significant. For these two parameters, the treatment with 10% (*v*/*v*) cinnamon oil determined significantly lower values if compared to the control at the same time of evaluation (Table 6).

Overall, it can be noted that there are small differences between the samples in relation to the type of filter paper inserted in the package (with or without 10% (*v*/*v*) cinnamon oil), but these differences do not appear to significantly change the characteristics of the fruit.

The pH values obtained on the strawberry samples are in line with the literature, where there is generally a decrease in organic acids used as an energy source in the ripening process [39,64]. However, the trends in terms of this process were found to be different according to the cultivar [65].

During storage, the process changes [66], and the data obtained in this study show a reduction in organic acids over time but also a decrease in total solids and a parallel increase the index of the catabolic processes within the fruit. 

The increase in moisture values for strawberries samples treated with EO is similar to that obtained by other authors using a coating on strawberries, where the presence of the EO limited gas exchanges and the breathing process [39]. The antioxidant properties of strawberries can be influenced by various external factors, including cultural systems (organic or not) or the storage temperature [67]. Jin et al. [68] evidenced that strawberries treated with EOs and stored at a higher temperature (>10°C) exhibited higher activity in terms of antioxidants enzymes and antioxidant capacities than those stored at refrigerated temperatures. This is in accord with the data obtained related to higher levels of phenolics, total antioxidant activity and anthocyans in the strawberries stored in the presence of cinnamon essential oil after four days.

For the raspberries, the soluble solids, titratable acidity and pH data obtained in the present study agree with the ranges reported in the literature for red raspberries (related both to conventional or organic fertilization) [69]. The water percentages also correspond to values for raspberries at the complete ripening stage, such as in Kobori et al. [70]. The levels of total phenols and anthocyanins are within the ranges reported by Bobinaite et al. [71] concerning two different cultivars harvested in Lithuania. These authors reported a strong correlation between the radical scavenging capacity (obtained with a DPPH test) and total phenolic and total ellagic acid. The contribution of total anthocyanins was minimal, in agreement with the results presented by Weber et al. [72]. The highest antioxidant activity was observed at the ripe maturity stage in raspberries [70].

In the study by Jin et al. [68], raspberries treated with single components of EOs had higher antioxidant enzyme activities and higher levels of phenolics and anthocyanins during storage at 10 °C for seven days. The presence of EOs delayed the development of decay in raspberries. An increase in the antioxidant capacity would reduce the physiological deterioration and enhance the resistance of tissue against microbial invasion, reducing the spoilage of the berry fruits.

## 4. Conclusions

In recent years, essential oils extracted from plant species have been considered as an efficient alternative to chemical preservatives to prolong the shelf life of susceptible, perishable fruits such as small red fruits (strawberries and raspberries) and fresh-cut products without a loss in their quality in terms of texture, taste and appearance. This study has demonstrated the antifungal and antibacterial activity of different EOs due to a large amount of bioactive components such as phenolic compounds, aldehydes, terpenes and terpenoids, as shown by GC-MS and HPLC analyses. As the primary experiment, the antibacterial activities of binary combinations of EOs were determined to assess the kind of interactions between EOs by measuring the FIC index. In the in vivo evaluation, the fresh and biological fruits were collected from a chemical- and pesticide-free farm and the post-harvest shelf lives of fruits without any prior infection with fungal and bacterial strains after four days of incubation at room temperature were investigated. Weight loss measurements and physicochemical analyses of the fruits revealed that using filter paper covered by cinnamon oil at a concentration of 10% (*v*/*v*) is more effecting when preserving strawberries compared to raspberries. These results show that this approach appears to be valuable when it comes to preserving susceptible fruits from physical and biological damages in the field and marketing conditions at room temperature without adding any extra fungicidal chemicals. Therefore, the method is worth studying in-depth to verify the impact of EOs on longer storage times and in refrigerated conditions, corresponding to conservation during the sale of fruits and their home storage at low temperatures. Additionally, further studies will be also necessary in order to evaluate the sensory properties of the fruits treated with EOs, with the aim of verifying consumer acceptability. 

## Figures and Tables

**Figure 1 foods-12-00332-f001:**
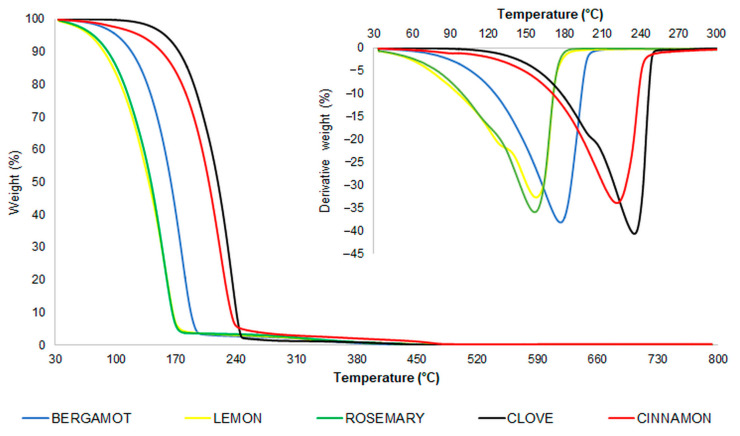
TG and DTG (in the inset) traces of bergamot, lemon, rosemary, clove and cinnamon oils, recorded in the range of 30–800 °C in an air-saturated atmosphere.

**Figure 2 foods-12-00332-f002:**
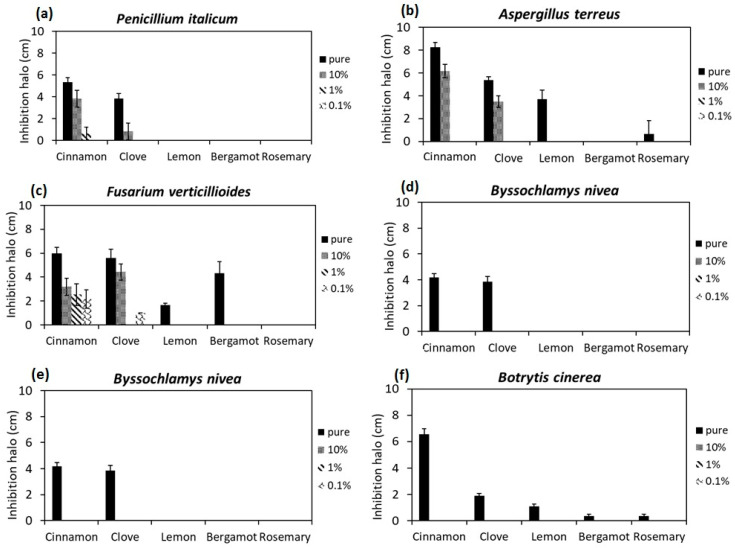
Antifungal activity evaluation of EOs with disc diffusion technique. Measurement of inhibition halo diameters (cm) created around the discs; (**a**) *Penicillium italicum*; (**b**) *Aspergillus terreus*; (**c**) *Fusarium verticillioides*; (**d**) *Byssochlamys nivea*; (**e**) *Rhizopus nigricans*; and (**f**) *Botrytis cinerea*. 10%, 1% and 0.1% indicate concentrations of EOs (*v*/*v*).

**Figure 3 foods-12-00332-f003:**
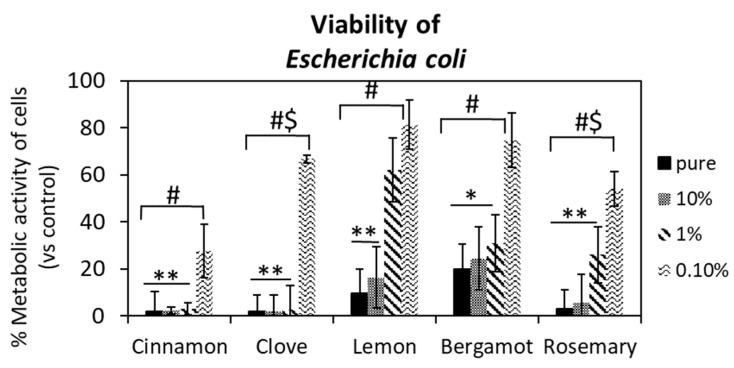
Metabolic activity of *E. coli* in the presence of EOs. The results were normalized with LB supplemented with 10% DMSO in PBS solution as a control sample. 10%, 1% and 0.1% indicate concentrations of EOs (*v*/*v*). * and ** indicate *p*-value < 0.05 and <0.01 between EO concentrations and control samples, respectively, while # and #$ show *p*-value < 0.05 and <0.01 between different concentrations of EOs, respectively.

**Figure 4 foods-12-00332-f004:**
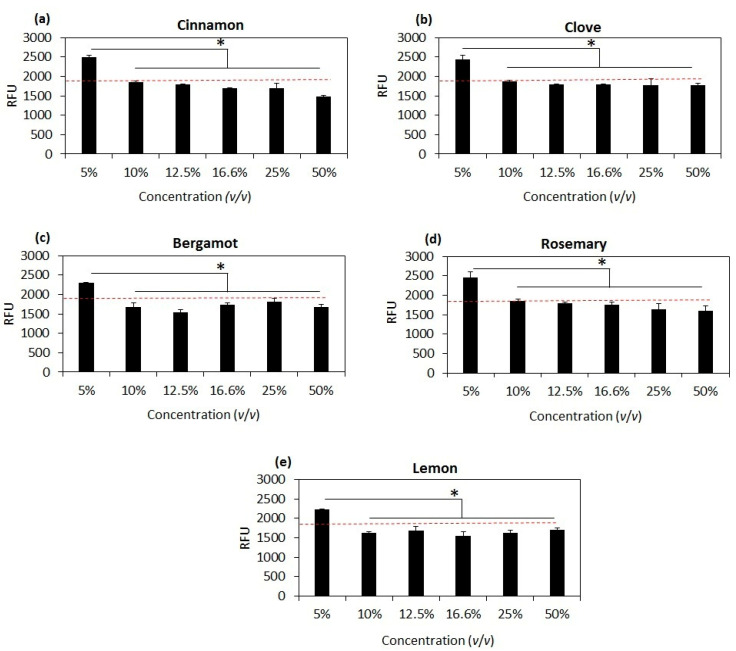
Minimum Inhibitory Concentration (MIC) of tested EOs by alamarBlue colorimetric assay; (**a**) cinnamon oil; (**b**) clove oil; (**c**) bergamot oil; (**d**) rosemary oil and (**e**) lemon oil; the red line indicates the Relative Fluorescent Unit (RFU) of the solution of 0.00015% alamarBlue in PBS as a control sample. * indicates *p*-value < 0.05.

**Figure 5 foods-12-00332-f005:**
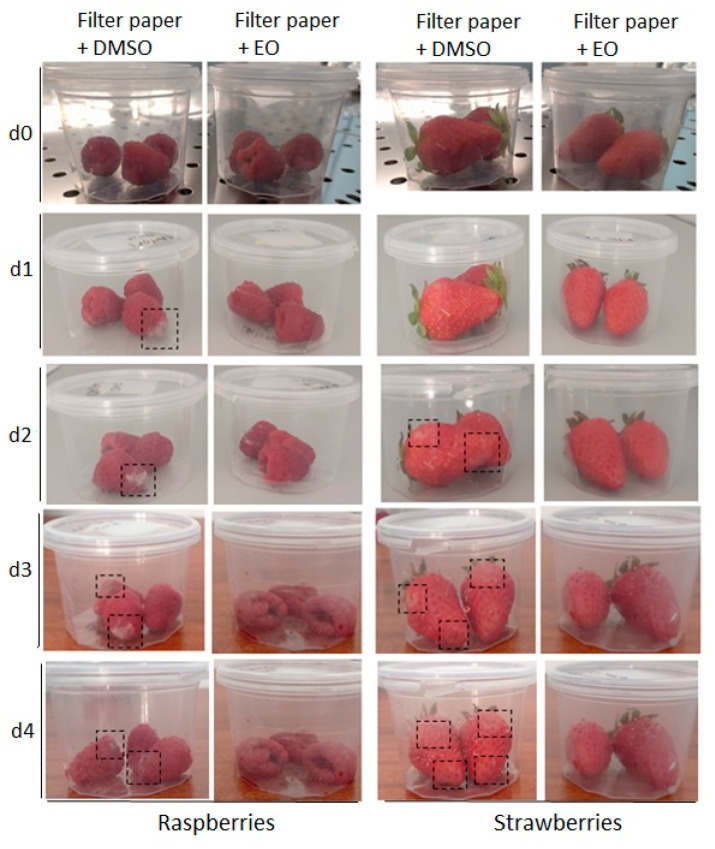
Raspberries and strawberries were incubated by sterile filter paper with 10% (*v*/*v*) cinnamon oil for four days (**d0**–**d4**) at room temperature. Sterile filter paper covered with 10% DMSO in PBS solution was used in the control sample.

**Table 1 foods-12-00332-t001:** Binary combinations of EOs with different concentrations and ratios.

Combined EOs	Concentration of Each EO (*v*/*v*)	Combination Ratio
Cinnamon + Clove	10% + 10%	50:50
Lemon + Cinnamon	10% + 10%	50:50
Bergamot + Cinnamon	10% + 10%	50:50
Cinnamon + Clove	5% + 5%	50:50
Lemon + Cinnamon	5% + 5%	50:50
Bergamot + Cinnamon	5% + 5%	50:50

**Table 2 foods-12-00332-t002:** Main components in bergamot, lemon, rosemary and cinnamon oils, obtained by GC-MS analysis. Components with relative abundance <5.0 % were not reported.

Essential Oil	Compound	Relative Abundance (%)
Bergamot	Limonene	39.5
	Linalyl acetate	23.9
	Linalool	10.2
Lemon	Limonene	63.7
	Pinene	9.2
	Lavandulyl-acetate	6.2
Rosemary	Eucalyptol	60.5
	Camphor	14.6
Cinnamon	Cinnamaldehyde	54.5
	Eugenol	10.2
	Caryophyllene	6.0

**Table 3 foods-12-00332-t003:** T_onset_ and T_peak_ of bergamot, lemon, rosemary, clove and cinnamon oils obtained by TGA.

Essential Oil	T_onset_ (°C)	T_peak_ (°C)
Bergamot	66.3	177.3
Lemon	40.6	156.8
Rosemary	44.3	131.6/158.6
Clove	126.0	200.6/234.9
Cinnamon	73.6	232.0

**Table 4 foods-12-00332-t004:** FIC indices of EOs combined.

Essential Oils (*v*/*v*)	FIC Index	Interpretation
Cinnamon + Bergamot (10%)	0.9	Additive
Cinnamon + Bergamot (5%)	1.58	Indifference
Cinnamon + Lemon (10%)	0.9	Additive
Cinnamon + Lemon (5%)	0.9	Additive
Cinnamon + Clove (10%)	1.6	Indifference
Cinnamon + Clove (5%)	1.5	Indifference

**Table 5 foods-12-00332-t005:** Chemical parameters quantified in the strawberries, during the four days of storage (C = filter paper covered with 10% DMSO in PBS solution as control sample, T = filter paper covered with 10% (*v*/*v*) cinnamon oil). Different letters represent significant differences within each row (*p* < 0.05) between the control samples. * indicates a significant difference between treated and control sample at the same time.

Strawberries	T0	T2C	T2T	T4C	T4T
Moisture (%)	89.1 ± 0.3 ^b^	90.70 ± 0.16 ^a^	91.10 ± 0.18 *	90.8 ± 0.6 ^a^	92.00 ± 0.18 *
Total Solids (%)	10.9 ± 0.3 ^a^	9.26 ± 0.16 ^b^	8.93 ± 0.18 *	9.2 ± 0.6 ^b^	7.96 ± 0.18 *
Total Soluble Solids (°Brix)	9.10 ± 0.07 ^a^	7.70 ± 0.12 ^b^	7.1 ± 0.4 *	7.5 ± 0.4 ^b^	6.95 ± 0.16
pH	4.28 ± 0.01 ^a^	4.03 ± 0.08 ^b^	3.92 ± 0.13	4.01 ± 0.02 ^b^	3.96 ± 0.04
Titratable Acidity (% citric acid)	0.636 ± 0.01 ^a^	0.68 ± 0.00 ^a^	0.69 ± 0.04	0.60 ± 0.07 ^a^	0.60 ± 0.12
Maturation Index (°BRIX/acidity)	14.23 ± 0.43 ^a^	11.41 ± 0.18 ^b^	10.2 ± 1.0 *	12.4 ± 1.1 ^a,b^	12 ± 3
Total Anthocyanins (mg C3GE/g s.s.)	0.13 ± 0.02 ^b^	0.43 ± 0.04 ^a^	0.40 ± 0.05	0.32 ± 0.08 ^a,b^	0.40 ± 0.13
Monomeric Anthocyanins (mg C3GE/g s.s.)	0.09 ± 0.02 ^b^	0.29 ± 0.04 ^a^	0.25 ± 0.05	0.29 ± 0.07 ^a^	0.35 ± 0.12
Total Phenolics (mg CE/g s.s.)	16.3 ± 0.6 ^a^	16.2 ± 0.9 ^a^	18.3 ± 0.9	16.5 ± 1.5 ^a^	20.1 ± 1.3 *
Total Antioxidant Activity (mg TE/g s.s.)	13.80 ± 0.08 ^a^	13 ± 2 ^a^	11.6 ± 1.0	13.4 ± 0.7 ^a^	16.0 ± 0.5 *

**Table 6 foods-12-00332-t006:** Chemical parameters quantified in the raspberries, during the four days of storage (C= filter paper covered with 10% DMSO in PBS solution as control sample, T = filter paper covered with 10% (*v*/*v*) cinnamon oil). Different letters represent significant differences within each row (*p* < 0.05) between the control samples. * indicates a significant difference between treated and control sample at the same time.

Raspberries	T0	T2C	T2T	T4C	T4T
Moisture (%)	84.3 ± 0.6 ^a^	85.1 ± 0.8 ^a^	85.30 ± 0.04	85.4 ± 0.4 ^a^	84.80 ± 0.10 *
Total Solids (%)	15.7 ± 0.6 ^a^	14.9 ± 0.8 ^a^	14.71 ± 0.04	14.6 ± 0.4 ^a^	15.25 ± 0.10 *
Total Soluble Solids (°Brix)	11.20 ± 0.00 ^c^	11.60 ± 0.00 ^a^	11.60 ± 0.00	11.40 ± 0.00 ^b^	11.40 ± 0.00
pH	3.43 ± 0.01 ^a^	3.43 ± 0.01 ^a^	3.42 ± 0.02	3.35 ± 0.02 ^b^	3.39 ± 0.01
Titratable Acidity (% citric acid)	2.08 ± 0.01 ^a^	1.78 ± 0.00 ^c^	1.88 ± 0.02 *	1.84 ± 0.01 ^b^	1.76 ± 0.00 *
Maturation Index (°BRIX/acidity)	5.38 ± 0.02 ^c^	6.51 ±0.01 ^a^	6.17 ± 0.06 *	6.19 ±0.05 ^b^	6.48 ± 0.00 *
Total Anthocyanins (mg C3GE/g s.s.)	1.25 ± 0.06 ^c^	2.45 ± 0.00 ^a^	1.89 ± 0.08 *	1.92 ± 0.07 ^b^	1.48 ± 0.03 *
Monomeric Anthocyanins (mg C3GE/g s.s.)	1.12 ± 0.07 ^c^	2.01 ± 0.01 ^a^	1.55 ± 0.06 *	1.6 ± 0.08 ^b^	1.23 ± 0.02 *
Total Phenolics (mg CE/g s.s.)	18.0 ± 0.8 ^a^	17.4 ± 0.5 ^a^	16.23 ± 0.03	17.31 ± 0.13 ^a^	17.65 ± 0.12
Total Antioxidant Activity (mg TE/g s.s.)	17.9 ± 0.6 ^a^	18.0 ± 1.0 ^a^	19.2 ± 0.7	16.2 ± 0.4 ^a^	16.7 ± 0.8

## Data Availability

The datasets generated for this study are available on request to the corresponding author.

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
