# Peer review of "Screening of Different Essential Oils Based on Their Physicochemical and Microbiological Properties to Preserve Red Fruits and Improve Their Shelf Life"

_foods, 2023, doi:10.3390/foods12020332_

Round 1

Reviewer 1 Report

This manuscript is on the analysis of some essential oil (EO) composition, antimicrobial activity and their application on the prolonging the shelf life of some fruits such as strawberries and raspberries. The subject of the study is interesting. However, there should be some revision and clarification particularly in method section.

Title of the manuscript should be revised to show the all aspects of the study and analysis.

In abstract, it should be clarified how EO diluted and were applied to fruits.

In method section, in preparation section,  the dilution are 100 and then 10 % and less. What about other levels, it would be reasonable to use 100, 90, 80 and so on, or 100, 75, 50 and 25 and so on. Coming down from  100 to 10% does not sound suitable.

There should be also some organoleptic test to show the consumer acceptance as some of these EO has high flavour and smell.

For Preparation of the phenolics extract, please give the reference and also, in most cases methanol and water used to extract, in this study ethanol was used to extraction.

What would be effect of EO on the weight loss? Some experiments could be ignored and deleted from manuscript as they are not affected by the EO.

In Table 2, reporting of the main components, generally about30- 40 % of the EO composition is missing? Also, if EO were from a reliable company, there was no need to analyse as their composition have already been analysed and published in many papers and books.

Line 617, while there are no variations regarding the acidity. If EO affected the shelf life and the mold content of the fruits, there should be differences in acidity and pH of the fruits as well due to the ability of mold and yeast on the hydrolysis and other reactions.

Line 621-627, stating totally controversial results and data.

Author Response

This manuscript is on the analysis of some essential oil (EO) composition, antimicrobial activity and their application on the prolonging the shelf life of some fruits such as strawberries and raspberries. The subject of the study is interesting. However, there should be some revision and clarification particularly in method section.

  1. Title of the manuscript should be revised to show the all aspects of the study and analysis.

The title was changed to “Screening of different essential oils based on their physicochemical and microbiological properties to preserve red fruits and improve their shelf-life”

  1. In abstract, it should be clarified how EO diluted and were applied to fruits.

Some phrases indicating the preparation of different concentrations of EOs and their use in in vivo condition were added in the abstract section to clarify the missed information.

  1. In method section, in preparation section, the dilution is 100 and then 10 % and less. What about other levels, it would be reasonable to use 100, 90, 80 and so on, or 100, 75, 50 and 25 and so on. Coming down from 100 to 10% does not sound suitable.

Thank you very much for this comment. These concentrations of essential oils were chosen according to previous literature indicating cytotoxic effect of pure essential oils; for instance, Azzimonti et al. (2015) investigated the human gingival fibroblast (HGF) and mucosal keratinocytes (HKF) viability at three different dilutions of EO (10-, 100- and 1000-fold). Their results revealed about 40% reduction of HGF and HKF viability at dilution 10-fold in compared to 100- and 1000-fold dilutions [1].

  1. There should be also some organoleptic test to show the consumer acceptance as some of these EO has high flavour and smell.

We totally agree with the reviewer, as essential oils, depending on their concentration, can modify some organoleptic traits of the foodstuff. Indeed, in a previous study, we showed that cheese slices treated with parsley essential oil were appreciated by the panellists [2]. We are aware of this issue, but the present study mainly focused on physicochemical and microbiological traits of the investigated essential oils. We have indeed demonstrated their thermal stability by thermogravimetric analysis, among other results, and effects on food flavor are beyond the scope of the present study. Our interest in these essential oils still remains and we will certainly investigate their effects on food flavor in further studies, as well as their application in food packaging.

  1. For Preparation of the phenolics extract, please give the reference and also, in most cases methanol and water used to extract, in this study ethanol was used to extraction.

The extraction of polyphenolic compounds was optimized in the laboratory, choosing the best conditions among different solvents (aqueous ethanol and methanol in different proportions); for this reason, a specific reference cannot be added. As stated by the reviewer, several papers use methanol in different concentrations for the phenolic extraction; sometimes the solvents are also acidified. However, in many cases aqueous ethanol has been used, in particular for anthocyanin extraction; for example refer to [3,4]. Furthermore, the aim was not to extract as many polyphenols as possible from the samples, but to evaluate the impact of EO and time on the shelf-life, so we didn’t necessarily choose the most efficient method, but the most practical and fastest one, as long as it was the same for all the samples.

  1. What would be effect of EO on the weight loss? Some experiments could be ignored and deleted from manuscript as they are not affected by the EO.

As suggested by the reviewer, the results on weight loss were deleted

  1. In Table 2, reporting of the main components, generally about30- 40 % of the EO composition is missing? Also, if EO were from a reliable company, there was no need to analyse as their composition have already been analysed and published in many papers and books.

Thank you for this comment. The main components in Table 2 represent 71-79% of the EOs composition. The missing percentages (about 20-30%) are due to other components showing relative abundances for each molecule <5%. The EOs purchased from Farmalabor have been analysed by the company, but in the technical datasheet only a range of relative abundance were reported. Therefore, the analysis of the EOs compositions was carried out to ascertain the composition that may change from batch to batch.

This part was added in the section “3.1 characterizations of essential oils”.

  1. Line 617, while there are no variations regarding the acidity. If EO affected the shelf life and the mold content of the fruits, there should be differences in acidity and pH of the fruits as well due to the ability of mold and yeast on the hydrolysis and other reactions.

In strawberries, no variation of the titratable acidity (% citric acid) was detected, while pH significantly decreased during storage in both control and treated fruits. Probably, the titratable acidity only partially contributes to the pH of the fruits, and the effect of EO on the change of pH is limited. Concerning raspberries, some significant differences among samples were observed for the titratable acidity; however, these differences are relevant mainly on the statistical point of view, and the impact on fruit properties could be considered as minimal. The same comment was added in the manuscript.

  1. Line 621-627, stating totally controversial results and data.

We thank the reviewer for the useful comment. There was an error (now corrected): the discussion is referred to the data reported in Table 6.

Refernces:

  1. Azzimonti, B.; Cochis, A.; Beyrouthy, M.E.; Iriti, M.; Uberti, F.; Sorrentino, R.; Landini, M.M.; Rimondini, L.; Varoni, E.M. Essential Oil from Berries of Lebanese Juniperus excelsa M. Bieb Displays Similar Antibacterial Activity to Chlorhexidine but Higher Cytocompatibility with Human Oral Primary Cells. Molecules 2015, 20, 9344-9357.
  2. Vitalini, S.; Nalbone, L.; Bernardi, C.; Iriti, M.; Costa, R.; Cicero, N.; Giarratana, F.; Vallone, L. Ginger and parsley essential oils: chemical composition, antimicrobial activity, and evaluation of their application in cheese preservation. Natural Product Research 2022, 1-6, doi:10.1080/14786419.2022.2125965.
  3. Mane, S.; Bremner, D.H.; Tziboula-Clarke, A.; Lemos, M.A. Effect of ultrasound on the extraction of total anthocyanins from Purple Majesty potato. Ultrasonics Sonochemistry 2015, 27, 509-514, doi:https://doi.org/10.1016/j.ultsonch.2015.06.021.
  4. Catena, S.; Rakotomanomana, N.; Zunin, P.; Boggia, R.; Turrini, F.; Chemat, F. Solubility study and intensification of extraction of phenolic and anthocyanin compounds from Oryza sativa L. ‘Violet Nori’. Ultrasonics Sonochemistry 2020, 68, 105231, doi:https://doi.org/10.1016/j.ultsonch.2020.105231.

Reviewer 2 Report

This article is about the use of essential oils to preserve red fruits and improve their shelf-life.

Although the topic was not so novel but they had evaluated many tests and the reported results were comprehensive and complete, and I think this article is acceptable with minor revised.

1-      The introduction section should be shortened and rewritten;

2-      Refer to more recent articles (since 2015) in this field (the effect of essential oil on increase shelf-life fruits like strawberry)

https://doi.org/10.1111/ijfs.15130, https://doi.org/10.17660/ActaHortic.2021.1327.79

Author Response

This article is about the use of essential oils to preserve red fruits and improve their shelf-life.

Although the topic was not so novel but they had evaluated many tests and the reported results were comprehensive and complete, and I think this article is acceptable with minor revised.

  1. The introduction section should be shortened and rewritten.

Following the suggestion of the reviewer, we have shortened and rephrased the introduction section.

  1. Refer to more recent articles (since 2015) in this field (the effect of essential oil on increase shelf-life fruits like strawberry)

https://doi.org/10.1111/ijfs.15130,

https://doi.org/10.17660/ActaHortic.2021.1327.79

           Following the suggestion of the reviewer, these references have been added in the main text.

Reviewer 3 Report

In this investigation, the application of essential oils to preserve red fruits and improve theier shelf-life was researched. This might be provide some useful informations for readers. Some revisions should be conducted.

1. The key result datas should be added in the abstract section.

2. The investigation of antibacterial activity, only the Escherichia coli  used, might not be enough. Some other bactierals should be also considered.

3. The introduction of red fruits should be in detail.

4. The application of EOs on the preservation of fruits researched by others should be concluded in the introduction section.

5. The result data of each indexes should be introduced in the results section but not only in the tables or figures.

6. The more related artilces should be added and discussed in the results section to explane the effect mechansim of EOs on the quality of fruits.

7. The English language should be improved in the whole manuscript.

Author Response

In this investigation, the application of essential oils to preserve red fruits and improve their shelf-life was researched. This might be provided some useful information for readers. Some revisions should be conducted.

  1. The key result data should be added in the abstract section.

In order to clarify the abstract, some more results have been added to this section.

  1. The investigation of antibacterial activity, only the Escherichia coli used, might not be enough. Some other bactierals should be also considered.

Thank you so much for pointing this out. A report of the Food and Drug Administration (FDA) published in 2020 indicated 48 million cases of foodborne illness yearly in the United States with an estimate of 128000 hospitalizations and 3000 deaths. Some common bacterial strains responsible for foodborne outbreaks are as follows: E. coli, Salmonella sp. and Listeria sp. [1]. Therefore, in this study similar to some previous literature [2,3], E. coli was selected as a representative of human foodborne pathogens. Our interest in these EOs still remains and as further experiments antibacterial evaluation of EOs against other foodborne pathogens will be analyzed.

  1. The introduction of red fruits should be in detail.

The following phrases about anti-inflammation properties of small, red fruits were added in the introduction section. “Therefore, it is believed that fresh fruits consumption increases the human body’s protection from various non-communicable diseases like neurological diseases, cardiovascular disease, diabetes mellitus obesity and some cancers [4]. Huang et al. (2016) reported that strawberries include a high volume of anti-inflammatory polyphenols which have been shown to attenuate meal-induced postprandial inflammation and oxidative stress [5]”.

  1. The application of EOs on the preservation of fruits researched by others should be concluded in the introduction section.

Thank you for this comment. As the reviewer suggested us, some studies about the EOs application on prolonging the fruits’ shelf-life were added; as following: “Some research was carried out to investigate the effect of the EOs on fruit decay, for instance Ansarifar et al. (2021) investigated strawberries quality preservation with Thyme EO encapsulated in zein nanofiber. Since the thymol oil was gradually released into the fruits’ packages, their physical-chemical properties remained constant and bacterial, fungal contaminations decreased after 15-day storage at 4 °C [6]. In addition, Guerreiro et al. (2021) reported that using edible coating with a combination of alginate (2%), citral (0.15%) and eugenol (0.1%) preserve strawberries with no changes in sensorial and nutritional properties up to 15 days [7]”.

  1. The result data of each indexes should be introduced in the results section but not only in the tables or figures.

As the reviewer suggested some phrases explaining the indexes have been added in the main text.

  1. The more related articles should be added and discussed in the results section to explain the effect mechanism of EOs on the quality of fruits.

Thank you for pointing this out. In the related section (section 3.5) the following phrases as discussion were added: “These results are in agreement with previous research; for instance, Piechowiak et al. (2022) showed that starch- and gelatin- based edible coatings doped with Cinnamon oil (10%) considerably declined the fungal growth on the blueberries 1.29 log cycles lower than ones in the control specimens after 10 days storage at 4 °C [8]. Rashid et al. (2020) investigated the storage stability of apples coated by different concentrations of Cinnamon oil at 5 °C for 2 months; based on physiochemical analysis, microbial assay and sensory evaluation during storage time it was revealed that 5% Cinnamon oil prevented the fruits spoilage with no variation in nutritional values in apples [9]”.

  1. The English language should be improved in the whole manuscript.

In the revised version, all the authors have gone through the manuscript carefully, and by using a grammar checking program, and corrected the errors. We now believe that we have significantly improved the language.

References:

  1. www.fda.gov. Available online: https://www.fda.gov/food/outbreaks-foodborne-illness/foodborne-pathogens. 2020.
  2. Zhang, J.; Ye, K.P.; Zhang, X.; Pan, D.D.; Sun, Y.Y.; Cao, J.X. Antibacterial Activity and Mechanism of Action of Black Pepper Essential Oil on Meat-Borne Escherichia coli. Front Microbiol 2017, 7, doi:10.3389/fmicb.2016.02094.
  3. Mouatcho, J.C.; Tzortzakis, N.; Soundy, P.; Sivakumar, D. Bio-sanitation treatment using essential oils against E. coli O157:H7 on fresh lettuce. New Zealand Journal of Crop and Horticultural Science 2017, 45, 165-174, doi:10.1080/01140671.2016.1269813.
  4. Cosme, F.; Pinto, T.; Aires, A.; Morais, M.C.; Bacelar, E.; Anjos, R.; Ferreira-Cardoso, J.; Oliveira, I.; Vilela, A.; Gonçalves, B. Red Fruits Composition and Their Health Benefits—A Review. Foods 2022, 11, 644.
  5. Huang, Y.; Park, E.; Edirisinghe, I.; Burton-Freeman, B.M. Maximizing the health effects of strawberry anthocyanins: understanding the influence of the consumption timing variable. Food & Function 2016, 7, 4745-4752, doi:10.1039/C6FO00995F.
  6. Ansarifar, E.; Moradinezhad, F. Preservation of strawberry fruit quality via the use of active packaging with encapsulated thyme essential oil in zein nanofiber film. International Journal of Food Science & Technology 2021, 56, 4239-4247, doi:https://doi.org/10.1111/ijfs.15130.
  7. Guerreiro, A.; Gago, C.; Miguel, M.G.; Faleiro, M.L.; Antunes, M.D. Improving the shelf-life of strawberry fruit with edible coatings enriched with essential oils. 2021; pp. 597-606.
  8. Piechowiak, T.; Grzelak-BÅ‚aszczyk, K.; Sójka, M.; Skóra, B.; Balawejder, M. Quality and antioxidant activity of highbush blueberry fruit coated with starch-based and gelatine-based film enriched with cinnamon oil. Food Control 2022, 138, 109015, doi:https://doi.org/10.1016/j.foodcont.2022.109015.
  9. Rashid, Z.; Khan, M.R.; Mubeen, R.; Hassan, A.; Saeed, F.; Afzaal, M. Exploring the effect of cinnamon essential oil to enhance the stability and safety of fresh apples. Journal of Food Processing and Preservation 2020, 44, e14926, doi:https://doi.org/10.1111/jfpp.14926.